# Joint Associations of Leisure Screen Time and Physical Activity with Academic Performance in a Sample of Japanese Children

**DOI:** 10.3390/ijerph17030757

**Published:** 2020-01-24

**Authors:** Kaori Ishii, Kenryu Aoyagi, Ai Shibata, Mohammad Javad Koohsari, Alison Carver, Koichiro Oka

**Affiliations:** 1Faculty of Sport Sciences, Waseda University, Saitama 359-1192, Japan; Javad.Koohsari@baker.edu.au (M.J.K.); koka@waseda.jp (K.O.); 2College of Economics, Kanto Gakuin University, Yokohama 236-8501, Japan; kenryu.aoyagi@gmail.com; 3Faculty of Health and Sport Sciences, University of Tsukuba, Ibaraki 305-8574, Japan; shibata.ai.ga@u.tsukuba.ac.jp; 4Behavioural Epidemiology Laboratory, Baker Heart and Diabetes Institute, Melbourne, VIC 3004, Australia; 5Melbourne School of Population and Global Health, The University of Melbourne, Melbourne, VIC 3010, Australia; 6Mary MacKillop Institute for Health Research, Australian Catholic University, Melbourne, VIC 3000, Australia; Alison.Carver@acu.edu.au

**Keywords:** sedentary behavior, screen time, active lifestyle, elementary school, academic outcome, children

## Abstract

Studies have shown the potential effects of sedentary behavior and physical activity on not only physical and mental health but also academic performance in children. Nevertheless, studies have only focused on either sedentary behavior or physical activity. Examining the joint effects of both behaviors on academic performance provides detailed insights into the patterns of these behaviors in relation to children’s academic achievement. The present study investigated the joint longitudinal associations of physical activity and screen time with academic performance among Japanese children. The screen time and physical activity of 261 children aged 7–10 years were assessed, and their academic performance was evaluated one year later. Multivariate logistic regression analysis was used to examine the joint associations of screen time and physical activity with academic performance adjusted for demographic characteristics. Children with low screen time and physical activity had 2.04 (95% confidence interval: 1.11–3.78) times greater odds of having high academic performance compared to children with high screen time and low physical activity, while children with low screen time and high physical activity had 2.75 (1.17–6.43) times greater odds (boys; 4.12 (1.19–14.24)). Low screen time was related to high academic performance after one year, regardless of the physical activity level.

## 1. Introduction

High levels of sedentary behavior and insufficient physical activity are associated with poor health such as obesity, cardiometabolic health risks, and mental health problems [1,2,3]. For children, several current physical activity guidelines [4,5] recommend no more than two hours per day of recreational screen time (i.e., watching TV, DVDs, or videos, playing TV games, and using computers or the Internet), limiting the amount of sedentary transportation (e.g., car travel), and avoiding prolonged periods of sitting. The World Health Organization also recommends that children and adolescents aged 6–17 years engage in at least 60 min of moderate-to-vigorous physical activity (MVPA) per day [6]. Nevertheless, in Japan, approximately 60% of children were found to exceed the two hours per day maximum of sedentary behavior [7]. Additionally, about half of the elementary school children in Japan did not achieve recommended physical activity levels [8]. Despite the accumulating evidence for the health risks of an inactive lifestyle, children continue to spend more time being sedentary and less time being physically active [9]. Therefore, reducing children’s sedentary behavior and increasing their physical activity have become key public health priorities.

Recent studies have shown the potential effects of sedentary behavior and physical activity on academic performance [3,10,11]. For example, a systematic review of 35 studies found spending more than two hours per day in front of a screen to be negatively associated with academic achievement among school-aged children [3]. Conversely, engaging in at least 60 min of physical activity per day for five or more days per week was associated with higher academic achievement [12]. Education level in childhood is a strong predictor of future wealth and health [13]. For example, in the Netherlands, highly educated individuals tend to live longer and have better health than individuals with less education [14]. Therefore, considering children’s academic performance may be important for addressing public health problems across their lifespan.

In most research examining the associations of sedentary behavior or physical activity with children’s academic performance, the focus has been solely on sedentary behavior or physical activity. Only a few previous studies [15] have explored how patterns of sedentary behavior and physical activity (such as low sedentary behavior and high physical activity, or low sedentary behavior and low physical activity) may affect academic performance in children. For example, Maher et al. [15] found that those who engaged in low MVPA, as well as low sedentary behavior, had lower academic performance, compared to those with high MVPA and low sedentary behavior, low MVPA and high sedentary behavior, and high MVPA and high sedentary behavior. They included objective measures of sedentary behavior and physical activity; however, time spent studying was considered as a part of sedentary behavior [15]. Objective measures of sedentary behavior have high reliability and validity [16], but they cannot distinguish between types of sedentary behavior, such as those that may benefit (e.g., studying time or reading) or harm (e.g., TV viewing or game use) academic performance. It is likely that these different types of sedentary behavior have varying effects on academic performance [17]. It is possible, therefore, that the inclusion of studying time in sedentary behavior may influence the observed associations in their study. In addition, to our knowledge, there is no study of physical activity and sedentary behavior in the context of Asia, with its unique cultural and social characteristics. Therefore, the present study examined the joint longitudinal associations of physical activity and leisure screen time (such as TV viewing and PC or game use) with academic performance in a sample of Japanese children.

## 2. Materials and Methods

### 2.1. Participants

This longitudinal study was conducted in a cohort of children attending 4 grades of an elementary school (aged 7–10 years) in Japan in 2016 and 2017. In total, 397 children (193 boys and 204 girls) in grades 2–5 completed the survey in 2016.

### 2.2. Instrumentation

#### 2.2.1. Screen Time and Physical Activity

Screen time and physical activity were assessed using a questionnaire in 2016. Screen time was divided into three domains [18]: (1) TV or video viewing, (2) video game use, and (3) Internet or e-mail use (on a computer or tablet) outside of school time. Participants were asked to report the average number of days per week and average time (hours and minutes) per day on weekdays and weekends during which they engaged in screen-based behaviors in each of the domains. The corresponding averages (in terms of frequency per week and minutes per day) were then multiplied and divided by seven to achieve the average total number of minutes per day of screen time. These data were then categorized using a scale currently employed in a national survey of Japanese students [18]. Screen time was dichotomized as ”low”(≤2 h) and “high” (>2 h), based on the guideline of the maximum of two hours per day [4,5].

Children’s physical activity was measured using a validated questionnaire [7,19] that asked participants whether they engaged in MVPA—that is, physical activity that induces harder than normal breathing, such as brisk walking, cycling, dancing, or swimming—for at least 60 min per day; and, if so, on how many days they engaged in it during the past week. Consistent with previous studies [8], children’s physical activity level was dichotomized as “low” (<60 min of MVPA per day) and “high” (≥60 min of MVPA per day).

#### 2.2.2. Academic Performance

Academic performance was assessed using the total grade point average of 4 school subjects in 2017. Each child was given a grade ranging from 1 (low) to 3 (high) by their teachers [20] in each of 8 school subjects (Japanese, mathematics, social studies, science, music, arts, home economics/vocational technology, and physical education). In this study, we used the average scores of Japanese, mathematics, social studies, and science, since these are commonly covered in the Japan entrance examinations to private junior high schools.

#### 2.2.3. Covariates

Data on children’s sex, grade, height, and weight in 2016 were used. Degree of obesity was calculated from the height and weight data. Definitions of weight status were based on those of the standard calculating formula in Japan, from the Ministry of Education, Culture, Sports, Science, and Technology [21]: [(measured weight − standard weight) ÷ standard weight] × 100(%). The degree of obesity was defined as follows: scores of +50 were considered a high level of obesity, +30 to +50 a moderate level, and +20 to +30 a low level; −20 to +20 were considered normal (i.e., a healthy weight range).

### 2.3. Procedure

The survey was conducted during the school term after obtaining permission to conduct the study from all the children’s parents or guardians. A previous study [22] suggested that children younger than 10 years of age are unable to accurately report their activity patterns. On the other hand, parental reports of physical activity among six-year-olds have been found to have a strong correlation with children’s heart rate measures during physical activity [23]. Therefore, the present study asked the parents or guardians of children in grades 2 and 3 to complete the questionnaire. They were instructed to answer the questionnaire at home and return it to the classroom teacher. Children in the fourth grade or above (over 10 years old) were asked to complete the questionnaire in the classroom, under the supervision of a teacher.

### 2.4. Data Analysis

A logistic regression analysis was performed to estimate the odds ratios (ORs) of high academic performance for the different physical activity and screen time groups. Analyses were conducted with the whole sample and then separately for boys and girls, adjusting for demographic characteristics. According to the cut-off points for screen time and physical activity, participants were classified into the following four categories: high screen time/low physical activity; low screen time/low physical activity; high screen time/high physical activity; low screen time/high physical activity. The high screen time/low physical activity group was used as the reference group. All statistical analyses were performed using SPSS Statistics 24.0J for Windows (IBM Corp., Armonk, NY, USA). A *p*-value of <0.05 was considered statistically significant.

## 3. Results

### 3.1. Participant Characteristics

Table 1 shows the demographic characteristics, physical activity, and academic performance data. A total of 261 children (127 boys and 134 girls) completed the survey (response rate: 65.7%). The mean value (–0.62) of the degree of obesity was normal. The average screen time per day and the average number of days that participants engaged in 60 min or the recommended amount of MVPA per week were 140.5 (82.8) min/day and 4.4 (2.0) days/week, respectively. The average score for academic performance was 1.3 (0.5). Many participants were in the high screen time/low physical activity group (44.4%), followed by the low screen time/low physical activity (32.6%), high screen time/high physical activity (11.9%), and low screen time/high physical activity groups (11.1%).

### 3.2. Association of Academic Performance with Physical Activity and Screen Time

Table 2 shows the results of the logistic regression analysis. In the analysis of the whole sample, children in the low screen time/low physical activity group had 2.04 (95% CI: 1.11–3.78, *p* < 0.05) times greater odds of having high academic performance after one year compared with those engaged in high screen time/low physical activity. Similarly, participants in the low screen time/high physical activity group had 2.75 (95% CI: 1.17–6.43, *p* < 0.05) times greater odds of having high academic performance compared with those in the high screen time/low physical activity group. Among boys, compared to the reference group, participants in the low screen time/high physical activity group had 4.12 (1.19–14.25, *p* < 0.05) times greater odds of having high academic performance. No significant associations were found among girls.

## 4. Discussion

The present study examined the joint associations of leisure screen time and physical activity with children’s academic performance. We found that Japanese children with lower screen time had 2.0–2.7 times greater odds of having high academic performance, regardless of their physical activity level. This indicates that screen time is significantly associated with academic performance independent of physical activity. Notably, the present findings appear to be inconsistent with those of a previous study examining the same association, which reported that children in the low MVPA/low sedentary behavior group achieved significantly lower academic performance than did children in any other group, and those in the high MVPA/low sedentary behavior group had lower scores than did those in the low MVPA/high sedentary behavior group [15]. This suggests that children who engaged in more sedentary behavior had higher academic performance, regardless of physical activity level, than those who engaged in less sedentary behavior. This is likely because Maher et al. [15] used a different measure of sedentary behavior: rather than focusing on leisure screen time, they included all types of sedentary behavior (e.g., reading, studying at school, or extracurricular study time).

The present study found significant associations of leisure screen time and physical activity with the odds of high academic performance in boys, but not in girls. Previous studies [24,25] have shown that these associations differ by gender. Morita et al. [24] found a significant association between excess time spent using electronic devices and academic achievement among boys, but no significant association existed among girls. In addition, a study that stratified data by grade (children aged 6–8 years, or grades 1 to 3) found that different types of academic achievement (e.g., reading frequency, reading comprehension, and arithmetic skills) were associated with screen time and MVPA among boys in all grades for reading frequency, in grade 1 for reading comprehension, and arithmetic skills, but in grade 2 for arithmetic skills only among girls [25]. Possible reasons for the gender differences may relate to differences in children’s preference for sedentary behavior or academic subjects. There are also potential gender differences in the preference for academic subjects that could explain the findings [26]. For example, one previous study found that the time spent engaging in different types of sedentary behavior differed according to gender: boys spent more time using video games, while girls spent more time reading, listening to music, doing homework, and engaged in car travel [7]. In contrast, García-Hermoso et al. [27] examined the associations of screen time and MVPA with academic achievement, according to gender; for both boys and girls, they found no significant association between excessive screen time (>2 h/day) and academic achievement, but there was a significant association between MVPA and academic achievement. Further research is needed to clarify the gender differences in these associations.

This study has some limitations. First, we only focused on one school; thus, the results may lack representativeness and may not be generalizable. To assess the representativeness of the sample of children, the prevalence of obesity among different age groups of children in the present study was compared with data from the Annual Report of School Health Statistics Research in 2018 [28].The prevalence rates of obesity among the participants in the present study were 6.3% and 8.4% in the national data for male participants and 6.7% and 6.8% for female participants, respectively. Second, the study relied on self-report by older children and parent proxy-report measures for younger children, which may introduce errors due to varying interpretations of the questions.

Despite these limitations, this was, to our knowledge, the first study to examine the joint effects of leisure screen time and physical activity with academic performance in a sample of Japanese children. Our findings suggest the importance of spending less time being sedentary and more time being physically active for high academic performance after one year. Thus, to maintain children’s academic performance, it is necessary to focus on not only increasing physical activity but also decreasing screen time, especially among boys.

## 5. Conclusions and Implications for Health

Children with lower leisure screen time had greater odds of high academic performance after one year, regardless of their physical activity level. However, these odds were even higher for those who engaged in high levels of physical activity as well as low screen time. These findings highlight the need for effective and sustainable strategies to promote physical activity as well as decrease leisure screen time to improve the academic performance of Japanese school-aged children.

The present results show that school-based health programs or physical education courses should provide information to guardians or parents about the risk of excess leisure screen time for academic performance. For example, some Western countries such as Australia hold classroom lessons specifically focused on reducing screen time [29]. Moreover, some studies have reported that, aside from child education, parental engagement via take-home materials [30], parent newsletters [31], and meetings, as well as regular contact with teachers and parent or sports organizations can be effective for reducing screen time [32]. Provision of education and skills to parents and children that aim to reduce children’s screen time and increase their physical activity may improve not only academic performance but also children’s health across the lifespan.

## Figures and Tables

**Table 1 ijerph-17-00757-t001:** Descriptive characteristics, physical activity data, and academic psychological performance stratified by sex.

	Total		Boys		Girls	
n (%)	261	(100.0)	127	(48.7)	134	(51.3)
Grade, n (%)						
2nd	51	(19.5)	21	(16.5)	30	(22.4)
3rd	68	(26.1)	35	(27.6)	33	(24.6)
4th	65	(24.9)	34	(26.8)	31	(23.1)
5th	77	(29.5)	37	(29.1)	40	(29.9)
Degree of obesity						
Mean ± SD	−0.6 ± 11.7		−0.1 ± 11.7		−1.1 ± 11.8	
ST, min/day						
Mean ± SD	140.5 ± 82.8		152.1 ± 94.1		129.6 ± 69.0	
Moderate-to-vigorous PA, Number of days/week						
Mean ± SD	4.4 ± 2.0		4.6 ± 2.0		4.2 ± 2.0	
Academic performance						
Mean ± SD	1.3 ± 0.5		1.3 ± 0.5		1.3 ± 0.5	
Combined categories of physical activity and screen time, n (%)						
High ST/low PA	116	(44.4)	59	(46.5)	57	(42.5)
Low ST/low PA	85	(32.6)	37	(29.1)	48	(35.8)
High ST/high PA	31	(11.9)	17	(13.4)	14	(10.4)
Low ST/high PA	29	(11.1)	14	(11.0)	15	(11.2)

SD, standard deviation; ST, screen time; PA, physical activity.

**Table 2 ijerph-17-00757-t002:** Adjusted odds ratios of academic performance by physical activity and screen time categories.

	Total ^a^		Boys ^b^		Girls ^b^	
	OR	95% CI	OR	95% CI	OR	95% CI
High ST/low PA	ref.		ref.		ref.	
Low ST/low PA	2.04	1.11–3.78 *	2.10	0.87–5.08	1.84	0.77–4.44
High ST/ high PA	0.96	0.39–2.38	0.61	0.15–2.43	1.59	0.45–5.58
Low ST/ high PA	2.75	1.17–6.43 *	4.12	1.19–14.25*	1.86	0.56–6.22

OR, odds ratio; CI, confidence intervals; PA, physical activity; ST, screen time; ^a^ Adjusted by sex, grade, and degree of obesity; ^b^ Adjusted by grade, and degree of obesity; * *p* < 0.05; *p*-values are based on the logistic regression analysis.

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
