# Peer review of "Joint Associations of Leisure Screen Time and Physical Activity with Academic Performance in a Sample of Japanese Children"

_ijerph, 2020, doi:10.3390/ijerph17030757_

Round 1

Reviewer 1 Report

Among the keywords I would insert the word "screen time" which is often present in the text and is fundamental for understanding the comparison with physical activity. Also, within the keywords I would write "children" instead of "kids".

The presentation of the results should be clearer and more detailed. In this way the conclusions will be more supported by the results.

Author Response

We are grateful for your insightful comments. We have revised the manuscript in accordance with your suggestions as detailed below:

Comment 1:

Among the keywords I would insert the word "screen time" which is often present in the text and is fundamental for understanding the comparison with physical activity. Also, within the keywords I would write "children" instead of "kids".

Response:

As suggested, we have revised the keywords.

Comment 2:

The presentation of the results should be clearer and more detailed. In this way the conclusions will be more supported by the results.

Response:

Thank you for the comment. We have expanded the Results section. We have also revised the Conclusion section, accordingly.

Reviewer 2 Report

Undoubtedly, the abstract of the article deserves significant changes. It is here that we should find at least a few sentences about the purpose of research, the research problem and research questions.Then, the most important results of the research carried out should be placed there, as synthetically as much as possible.

The chapter 2.4. Human subjects approval statement should be moved to the footness. Also the remarks on SPSS Statistics should be moved there.

In the Chapter 2.1 Participants, the Authors should assess to what extent the sample selection of children aged 7-10 years - pupils of grades 1-4 of elementary school is representative for kids of that age in Japan

Author Response

Thank you for your insightful comments. We have addressed all the comments as detailed below:

Comment 1:

Undoubtedly, the abstract of the article deserves significant changes. It is here that we should find at least a few sentences about the purpose of research, the research problem and research questions. Then, the most important results of the research carried out should be placed there, as synthetically as much as possible.

Response:

Thank you. As suggested, we have significantly revised the Abstract.

Comment 2:

The chapter 2.4. Human subjects approval statement should be moved to the footness. Also the remarks on SPSS Statistics should be moved there.

Response:

Thank you. As suggested, we have moved these statements to the footnote.

Comment 3:

In the Chapter 2.1 Participants, the Authors should assess to what extent the sample selection of children aged 7-10 years - pupils of grades 1-4 of elementary school is representative for kids of that age in Japan

Response:

Thank you for this excellent comment. As per your suggestion, we have added information about the representativeness of the sample for children of those age groups in Japan in the limitations section.
